# Beneficial Microorganisms as Bioprotectants against Foliar Diseases of Cereals: A Review

**DOI:** 10.3390/plants12244162

**Published:** 2023-12-14

**Authors:** Ilham Dehbi, Oussama Achemrk, Rachid Ezzouggari, Moussa El Jarroudi, Fouad Mokrini, Ikram Legrifi, Zineb Belabess, Salah-Eddine Laasli, Hamid Mazouz, Rachid Lahlali

**Affiliations:** 1Phytopathology Unit, Department of Plant Protection, Ecole National of Agriculture Meknes, Km10, Rte Haj Kaddour, BP S/40, Meknes 50001, Morocco; il.dehbi@edu.umi.ac.ma (I.D.); oachemrk@enameknes.ac.ma (O.A.); ezzouggarirachid@gmail.com (R.E.); ikramlegr@gmail.com (I.L.); laaslisalaheddine@gmail.com (S.-E.L.); 2Laboratory of Plant Biotechnology and Molecular Biology, Faculty of Sciences, Moulay Ismail University, BP 11201, Zitoune, Meknes 50000, Morocco; h.mazouz@fs-umi.ac.ma; 3Laboratory of Biotechnology, Conservation, and Valorization of Natural Resources (LBCVNR), Faculty of Sciences Dhar El Mehraz, Sidi Mohamed Ben Abdallah University, Fez 30000, Morocco; 4Department of Environmental Sciences and Management, SPHERES Research Unit, University of Liège, 6700 Arlon, Belgium; meljarroudi@uliege.be; 5Biotechnology Unit, Regional Center of Agricultural Research, INRA–Morocco, Rabat 10080, Morocco; fmokrini.inra@gmail.com; 6Plant Protection Laboratory, Regional Center of Agricultural Research of Meknes, National Institute of Agricultural Research, Km 13, Route Haj Kaddour, BP 578, Meknes 50001, Morocco; zineb.belabess@inra.ma

**Keywords:** leaf cereal diseases, wheat, rice, maize, biocontrol

## Abstract

Cereal production plays a major role in both animal and human diets throughout the world. However, cereal crops are vulnerable to attacks by fungal pathogens on the foliage, disrupting their biological cycle and photosynthesis, which can reduce yields by 15–20% or even 60%. Consumers are concerned about the excessive use of synthetic pesticides given their harmful effects on human health and the environment. As a result, the search for alternative solutions to protect crops has attracted the interest of scientists around the world. Among these solutions, biological control using beneficial microorganisms has taken on considerable importance, and several biological control agents (BCAs) have been studied, including species belonging to the genera *Bacillus*, *Pseudomonas*, *Streptomyces*, *Trichoderma*, *Cladosporium*, and *Epicoccum*, most of which include plants of growth-promoting rhizobacteria (PGPRs). *Bacillus* has proved to be a broad-spectrum agent against these leaf cereal diseases. Interaction between plant and beneficial agents occurs as direct mycoparasitism or hyperparasitism by a mixed pathway via the secretion of lytic enzymes, growth enzymes, and antibiotics, or by an indirect interaction involving competition for nutrients or space and the induction of host resistance (systemic acquired resistance (SAR) or induced systemic resistance (ISR) pathway). We mainly demonstrate the role of BCAs in the defense against fungal diseases of cereal leaves. To enhance a solution-based crop protection approach, it is also important to understand the mechanism of action of BCAs/molecules/plants. Research in the field of preventing cereal diseases is still ongoing.

## 1. Introduction

Cereals, which include wheat (*Triticum* spp.), rice (*Oryza sativa* L.), and maize (*Zea mays* L.), make up the bigger part of crop production and account for 90% of the world’s cereal production [1]. Since they were originally domesticated thousands of years ago [2], they have been the primary source of nutrition for humans [2,3], accounting for more than 56% of daily calories and 50% of daily protein. Wheat is among the most widely grown small grains in the world [4]. Half of the world’s wheat is produced by the top five producers, which are France, Russia, China, India, and the United States [5]. However, these cultivated plants are subject to leaf diseases caused by pathogenic fungi (ascomycetes, basidiomycetes, etc.). They can be classified as biotrophic, necrotrophic, and hemibiotrophic based on their trophic biology [6], thus establishing a long-term relationship with the host and living inside the cellular plant. They are highly specialized and can absorb nutrients from plant cells causing cell death [7].

Cereal leaf diseases have been managed by eliminating alternative hosts [8], tillage as a cultural practice, and selecting resistant cultivars [9,10]. Biological and chemical control is also a key tool for reducing the severity of the disease and minimizing yield losses [11]. Although fungicides provide selective pressure that favors the evolution of fungicide-resistant plant pathogens [12]. Due to the high cost of fungicide treatment and the environmental and health risks posed by weather conditions, the adoption of genetic resistance has long been the preferred strategy [13]. For this reason, stakeholders need to develop a variety of management measures that rely mainly on chemical, biological, genetic, and agronomic aspects to ensure effective crop growth [14].

Currently, biocontrol products account for around 5% of the world market for crop protection (worth approximately USD 3 billion), and it is expected that by 2025, biocontrol products will grow at an annual rate of 8.84%, representing above 7% of the global crop protection market (valued at more than USD 4.5 billion) [15,16]. The use of antagonistic biocontrol agents (BCAs)—specifically, species belonging to the genera *Bacillus*, *Pseudomonas*, *Streptomyces*, *Cladosporium*, *Epicoccum*, and *Trichoderma*—against cereal foliar disease pathogens like species belonging to the genera *Puccinia*, *Septoria*, *Blumeria*, *Pyrenophora*, and *Bipolaris* has been widely reported. Many of these BCAs not only inhibit pathogen development but also directly stimulate plant growth [15].

In this paper, we review the essential alternatives to chemical applications for controlling foliar cereal diseases using BCAs as a good choice for disease prevention as well as their interactions with the plant and their mechanisms of action.

In this work, we retrieved bibliometric data on the biocontrol of the leaf diseases of cereals by using the SCOPUS database of which we chose the specific keywords to carry out this operation: “Cereals” or “Wheat” or “Rice” or “Maize” or “Biological Control Agents” or “Leaf diseases”. The VOS viewer (v1.6.9., Leiden University, Leiden, The Netherlands) processing software was used to create the bibliometric analysis. The study displays the distribution of the most pertinent publications regarding the biocontrol of diseases infecting cereal leaves by advantageous microorganisms. The results of the network analysis show a relationship between the keywords discovered and the general perspective of current investigations in the field (Figure 1).

## 2. The Main Foliar Diseases Affecting Cereals

### 2.1. Fungal Leaf Diseases of Wheat

Rusts are fungal diseases of higher plants including cereals. The pathogens responsible are obligate parasitic basidiomycetes of the genus *Puccinia* that colonize, grow, and reproduce only in living plant tissue, affecting photosynthesis by reducing leaf area and increasing the plant’s transpiration rate [17]. This leads to large incidents and significant yield losses under favorable environmental conditions [18,19,20]. There are two main wheat leaf rusts: brown rust (*Puccinia triticina)* and yellow rust (*Puccinia striiformis* f. sp. *tritici*). These pathogens have a dispersal capacity and short development cycles that enable them to spread rapidly in cultivated fields and induce explosive epidemics [20] (Figure 2).

Yellow rust (Yr) caused by *P. striiformis* a biotrophic wheat pathogen, is considered to be one of the principal threats to wheat production in past centuries [21,22]. It can reduce yields by 10–70% [23] if the wheat cultivar is susceptible and weather conditions are conducive to disease development [24,25]. Sudden epidemics in cultivars that were previously thought to be resistant have been produced by the *P. striiformis* pathogen’s dynamics to develop current races, and especially since 2010, the discovery and quick dissemination of current destructive races has caused catastrophic losses [26,27]. *P. striiformi* alternates between separate hosts during its life cycle; it lives on grasses during the asexual phase and on *Berberis* spp. during the sexual phase [28]. The life cycle of *P. striiformis* comprises five species of spores: urediniospores, basidiospores, and teliospores on wheat; pycniospores and aeciospores on *Berberis* spp. Urediniospores and teliospores are dicaryotic and teliospores generate haploid basidiospores [23]. The pycnial and aecidial spore stages have only recently been discovered [28]. When the supply of nutrients from infected tissues diminishes, the telial phase begins. As the urediniospores germinate, the germ tubes develop and penetrate the leaf stomata [29], triggering the stomatal cavity’s main infection hyphal development. When the hypha reaches the mesophyll or epidermal cells, the mother cells of the haustorium are formed. The haustorium then settles between the host cell’s plasma membrane and its cell wall [30]. The haustorium feeds the fungus by absorbing nutrients and water from the host plant cells [31]. Most of each haustorium is located in leaf mesophyll cells, but some are present in the leaf epidermal layer [32]. Secondary infection occurs from the hyphae of primary infection. These hyphae grow within the mesophyll cells’ intracellular area and branch the mesophyll layer to create a dense mycelial network. About a week after infection, the first bands of pustules begin to appear, and chlorotic spots are seen on the leaf surface. Ten to fourteen days after infection, sporulated pustules appear through the leaf epidermis with characteristic yellow spores. On adult plants, once the pustules become visible on the leaf surface, they appear in characteristic bands, as the leaves of adult plants, unlike young plants, have well-developed vascular bundles [33].

The wheat leaf rust (Lr) pathogen is renamed and referred to as *Puccinia triticina Eriks*. A healthy plant must go through several stages of growth before becoming affected by a disease. These steps are as follows: a pathogen enters the host, attaches to the host, recognizes the host, produces an aerosol, penetrates the host, infects the host, colonizes the host, and spreads, frequently by water and/or air. The five spore phases produced by the causal agent of leaf rust are urediniospores, teliospores, basidiospores (on the primary host), pycniospores, and aeciospores (on secondary hosts). The fungus produces spores repeatedly during the growth season because it is macrocyclic. According to climatic factors, host age, and genotype, fresh infections may happen every 7 to 10 days since each new spore can reinfect wheat [34]. Leaf rust can form ovoid to circular pustules on the adaxial and abaxial surfaces of leaves, with a diameter of up to 1.5 mm [34]. When leaf rust infections take place during the development stage from heading to senescence, around 20,000 spores can be generated per pustule [34]. The urediniospores germinate to form a germ tube after they touch down on a receptive host. According to Roelfs et al. [35], optimal spore germination takes place between 15 and 20 °C with constant dew for four to eight hours. It will move the germ tube over the leaf’s surface toward the stomata after germination [34]. The protoplasm of the germ tube accumulates at the tip of the hypha and forms a membrane when a stomata is present [34]. As a result of the appressorium, the stomata shut, allowing a penetrating peg to push through and into the substomatal region of the host [34]. A sub-stomatal vesicle is created by the penetrating dowel hydra. The substomatal vesicle interacting with the mesophilic cell is the source of invasion hyphae and the mother cell haustorial [34]. Nutrients can be transferred from the host to the fungus thanks to a strong interaction between their membranes [36].

*Septoria tritici* blotch (STB), caused by the fungus *Zymoseptoria tritici*, is currently the most detrimental disease of leaves affecting wheat crops, particularly in areas with favorable climatic conditions like Western Europe [37]. During severe epidemics, wheat crops may decrease considerably by 50% [38]. In the asymptomatic phase of a hemibiotrophic fungus, fungus hyphae develop between the cells of the leaf’s mesophyll without inducing host necrosis. A brief necrotrophic stage that lasts for approximately a week follows and is marked by a rise in fungal biomass, the synthesis of cell wall-degrading enzymes, the emergence of necrosis, and the growth of pycnidia [38,39]. Around 10 days after the infection process begins, the switch from biotrophy to necrotrophy happens rapidly [39], but this brief period might depend on the infected cultivar and natural circumstances [40]. As most wheat varieties lack significant host resistance to (STB), the disease is mainly controlled by the application of conventional fungicides. However, it commonly develops resistance to fungicidal substances and bypasses host resistance; the effectiveness of genetic and chemical control measures is generally threatened in the field [41,42], as well as its frequency of sexual reproduction and genetic recombination, which affect its high level of biological fitness [15,43,44].

One of the largest threats to the majority of cereal crops, including barley, oats, wheat, and triticale, is powdery mildew, which is caused by the biotrophic obligatory fungus *Blumeria graminis* [45]. It can only finish its life cycle on a live host and does not grow in axenic crops. *B. graminis* has a finely synchronized, strictly controlled asexual life cycle [46]. Ascospores and conidia serve a considerable role in the pathogenesis of the fungi that cause powdery mildew. As the disease progresses, *B. graminis* primarily reproduces asexually by perpetually producing conidia [46]. *B. graminis* conidia and ascospores germinate once they touch an appropriate surface of a host leaf, and they subsequently develop the appressorium and the infection structure to pierce their hosts [47]. Conidia with young conidiophores can act as an inoculum for the infection of volunteer plants produced by conidia and ascospore colonies after successful infection and haustorium development [48]. Therefore, conidia production and dissemination are essential to *B. graminis* pathogenesis [49]. The epidemic development of powdery mildew is highly influenced by the cultivars’ resistance and the impact of applied fungicides [24]. The disease has recently expanded more significantly in warmer, drier areas where agriculture practices have intensified due to increased irrigation, higher sowing rates, and the usage of nitrogen fertilizers [50], widely dispersed in regions with cool and dry temperatures. Although larger losses have been reported in other places, powdery mildew-related losses to wheat output in Western Europe are typically less than 10% [37], which may be due to the low seeding rates used, or when infection occurs very early and results in the death of individual tillers or whole plants. Historically, powdery mildew has largely been controlled by race-specific resistance genes, while cultivars have often been shown to be dependent on the emergence of new virulent races. *B. graminis* has a well-known and ubiquitous fungicide resistance problem [51]. 

One of the principal diseases of bread wheat is tan spot, often referred to as yellow spot, which is brought on by the fungus *Pyrenophora tritici-repentis*. Wheat leaves are affected by this disease, which results in chlorotic patches and necrotic lesions. The result is a reduction in the photosynthetic surface of the plant, which eventually causes leaf mortality and a decline in leaf quality [52,53]. Small oval- to diamond-shaped spots that appear periodically on wheat and triticale leaves are the symptoms of the leaves [54]. This disease may destroy plant vegetative parts like leaves, causing the infected plants to die before heading [55]. *P. tritici-repentis* conidia have a significant genetic diversity that has a good impact on host range and virulence [56,57]. They can also live for extended periods on plant debris and traverse great distances [58]. According to Lamari and Strelkov [59], yield losses might be as high as 50%. In all of the main wheat-growing countries of Syria, Brazil, Argentina, Australia, Algeria, the United States, Russia, Canada, and a few other nations, the brown-black leaf spot is a common occurrence [57,60,61]. ToxA, ToxB, and ToxC are the three known effectors produced by tan spots of wheat [54,62,63,64]. While ToxC is yet unidentified and considered to be a metabolite, ToxA and ToxB are proteins [54,64]. Biological control techniques, adequate agricultural practices, and developing resistant cultivars can all be used to manage disease in the field [54].

Spot disease is a common disease of wheat on all continents [65,66,67,68]. *Bipolaris sorokiniana* is the causal agent of wheat spot disease [65,69,70]. Losses are high, particularly in warmer parts of the world [71], including South Asia [72,73]. Leaf brown lesions represent the spot symptoms with a yellow halo that spreads over time to cover a larger area of the leaf. Lesions may be olive-brown, particularly under humidity, which favors fungal spore formation [67,68,74]. *B. sorokiniana* leaf infection can come through seeds, roots, or the atmosphere. The stomata of the hypocotyl can become infected if the pathogen is present in the soil, and the fungus can then spread to the roots, shoots, and coleoptiles [71]. *B. sorokiniana* may penetrate the cuticle and stomata, and its spores can germinate in 4 to 6 h [71,75].

### 2.2. Fungal Leaf Diseases of Rice

*Magnaporthe oryzae*, also known as *Pyricularia oryzae*, is a hemibiotrophic fungal disease that causes rice blast in staple crops including rice, millet, and barley. It may infect more than 50 different grass species [76]. The fungus infects the leaves, stems, nodes, panicles, and roots of rice plants at all phases of growth. The infection process begins when a conidium touches the cuticle of a rice leaf and adheres there [77]. In every part of the world where rice is grown, rice blast annually results in harvest losses of about 6% [78,79]. A conidium’s landing and adhesion to the rice leaf cuticle initiates the infection process. The germination conidium secretes an adhesive that facilitates adhesion to the cuticle. In favorable circumstances, the conidium germinates to form a germ tube, which then develops into an appressorium. The melanin layer that separates the appressorium’s cell wall from the cell membrane is unique, and the cell wall is differentiated. This layer facilitates the production of turgor pressure, which is then converted by the penetration peg into mechanical force and aids in penetrating the leaf cuticle. After entering the cell, hyphae proliferate quickly, causing disease and obvious blast symptoms [77]. 

*Bipolaris oryzae* (teleomorph: *Cochliobolus miyabeanus*), which causes rice brown spot (RBS), is a significant disease for rice around the world [80,81]. The most typical surviving structures and key inoculum sources are conidia and mycelia on seeds and in crop residues. The fungus affects grain hulls, panicles, glumes, stems, and sheaths [81]. Conidia and mycelia are considered to be the most prevalent survival structures and key inoculum sources on seeds and in crop leftovers. This fungal infection occurs in the seedling stage of rice and causes the majority of symptoms, weakening the plants and lowering grain yield [81]. Brown spotting symptoms begin to appear on both young and old leaves during the seedling stage. Less tillering results from a reduction in nutritional absorption and photosynthesis due to smaller leaves; conversely, a later stage of infection may cause less grain filling, as well as discolored, spotted, and shriveled grains. It is often referred to as poor farmers’ disease because it causes significant harm in the cool summer months, particularly on nutrient-deficient soils. The outcomes of the experiment demonstrated that environmental conditions directly impacted the spread of brown spot disease and the pathogen’s persistence in seed and soil. The relative humidity and temperature at which seeds are stored have an impact on the pathogen’s capacity to survive [82].

### 2.3. Fungal Leaf Diseases of Maize

*Exserohilum turcicum* (syn. *Setosphaeria turcica*), a hemibiotrophic fungus including in the class Dothideomycetes [83,84], is the pathogen that causes northern leaf blight (NLB) of maize (*Zea mays* L.), which is a widespread disease in maize-growing regions worldwide [85,86]. Losses in yield that vary from 15% to 30% [84] can mostly be attributed to the decrease in photosynthetic leaf surface caused by leaf wilting [87,88]. Tiny chlorotic patches that appear on the leaves following infection are among the earliest symptoms of NLB [89,90]. When an infection develops in the upper leaves through grain filling before silking, the condition becomes worse [91,92]. Conidia are the primary inoculum by which the disease spreads from one plant to another. It initially lives as a biotroph, feeding off living host tissue; subsequently, it switches to a necrotrophic lifestyle, killing the infected cells of the host [84].

The ascomycete *Cochliobolus heterostrophus* (Southern corn leaf blight, SCLB) is a common fungus distributed in maize planting areas worldwide [93]. A necrotrophic fungal infection called *Cochliobolus heterostrophus*, teleomorph (Nisikado) *Helminthosporium maydis* is the cause of SCLB disease [94,95]. Most often, the disease is discovered in humid corn-growing regions throughout the summer [96]. According to Feng et al. [97], maize leaf blight is responsible for 20–30% of maize output losses. The maize plant can become infected by both asexual and sexual ascospores throughout the disease’s polycyclic life cycle [96]. The conidies are released from the maize-infected lesions under hot and humid circumstances, and they spread to nearby plantules by wind or raindrops [98,99]. After the conidies appear on the leaf of a healthy plant, they develop on the leaf’s tissue by the process of producing germination tubes that can enter through the leaf’s tissue or through natural openings like stomates or hydathode to cause infection [96].

Grey leaf spot (GLS), induced by *Cercospora* spp., is one of the major diseases that impact the production of maize widely [100]. The disease was first reported in 1925 and has spread to the United States, China, Africa, Brazil, and Nepal [100,101] and currently represents a considerable threat to maize production [102]. Wet, dark, and tiny spots surrounded by a yellowish color are the first signs of GLS [103]. *C. zeae-maydis* generates conidia in debris in the late spring, and these conidia are propagated by rain or wind to infect fresh maize crops [103,104]. On the underside of young leaves, where conidia are generated on the leaf surface and enter via the stomata through a flattened hyphal structure known as the appressorium, primary inoculation occurs. The introduction of resistant hybrids is the most economical and ecologically responsible method of managing this disease [102,105]. The introduction of resistance alleles into the genetic resources of breeding programs is made possible by marker-assisted selection (MAS) [102,106]. Improving GLS resistance in maize breeding techniques requires the rapid discovery of GLS-resistant quantitative loci (QTL)/genes [102,107].

**Figure 2 plants-12-04162-f002:**
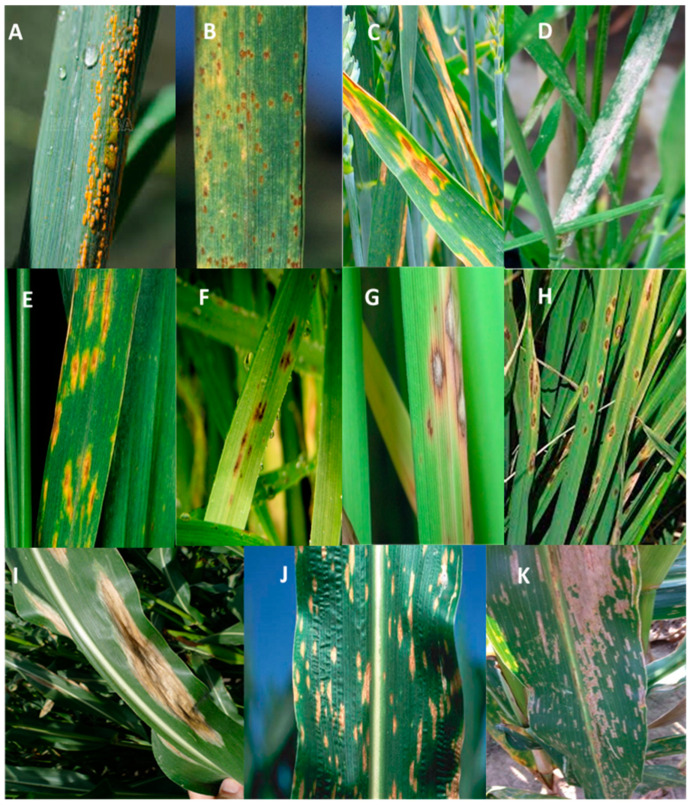
Main foliar diseases of cereals: Yellow rust of wheat (**A**) [108]; Leaf rust of wheat (**B**) [109], STB of wheat (**C**) [110]; Powdery mildew of wheat (**D**) [51]; Tan spot of bread wheat (**E**) [111]; *Bipolaris sorokiniana*; spot disease of wheat (**F**) [112]; Rice blast (**G**) [113]; Rice brown spot (**H**) [114]; Northern leaf blight of maize (**I**) [115]; Southern corn leaf blight (**J**) [116]; Grey leaf spot of maize (**K**) [117].

## 3. Current Control Strategies for Cereal Leaf Pathogens

### 3.1. Chemical Control

It is known that several fungicides, including those from the chemical families DMI (triazoles), Qols (strobilurins), SDHI (succinate dehydrogenase inhibitors), and chloronitriles (including chlorothalonil), are effective against diseases on cereal leaves. Reduced rates can be used early on to effectively manage stripe rust; however, late treatments are less cost-effective [118]. Triadimefon was widely utilized in China and North America to combat yellow rust [119]. Several different fungicide molecules—Triadimefon [19], Propiconazole [120], Fluoxastrobin, Tebuconazole, Propiconazole + Trifloxystrobin, Azoxystrobin, Azoxystrobin + Propiconazole, Prothioconazole + Tebuconazole [118,121], and Azoxystrobin + Flutriafol [122]—have recently been registered and shown to be very effective in controlling rusts and fungal leaf spots worldwide. *In planta* coumoxystrobin, a new fungicide based on strobilurin, has shown a good inhibitory effect on the mycelial growth of *M. oryzae* [123]. The use of fungicides is still controversial due to their increased costs, harmful impacts on the environment, and risks to community health despite their effectiveness in managing disease and safeguarding production. Due to decades of extensive fungicide usage, the likelihood of resistance emerging cannot be completely ruled out [124,125].

### 3.2. Management of Foliar Diseases Using Antagonists

Various national action plans in Europe have been drawn up to reduce the application of traditional synthetic fungicides in farming and actively support the development of environmentally and less harmful plant protection agents [126]. BCAs, commonly known as bio fungicides, are a viable alternative strategy for cereal pests and protection, fostering sustainable agriculture practices through their environmentally friendly use. BCAs exert a direct antagonistic impact on the pathogen through parasitism, antibiosis, or competition, achieved by secreting natural compounds. They also exert their biocontrol activity indirectly by induced systemic resistance (ISR) in cereals. Widely studied for its potential benefits, this approach aims to reduce reliance on chemical substances for plant protection, effectively controlling cereal diseases [127,128] (Figure 3). 

While BCAs are recognized for their effectiveness in managing pathogenic species, certain essential aspects of their functioning remain delicate and susceptible to various influences. Factors such as the growth and survival of BCAs are particularly influenced by natural conditions, including moisture, temperature, and nutrient availability, as well as application timing (curative vs. preventative) [129]. The interplay of these elements, along with the surrounding microbial environment, the targeted pathogen, and the intrinsic characteristics of the biocontrol agent itself, contributes to the intricate dynamics of biological control [130]. Ultimately, optimal outcomes are more likely to be achieved when favorable conditions are met, especially through the synergistic use of diverse BCAs, such as combinations of bacterial and fungal species. This approach is particularly advantageous when the modes of action of these agents are complementary [131,132].

#### 3.2.1. Biocontrol with *Bacillus* spp. as BCAs

*Bacillus* spp., one of the frequently endophytic bacteria, garnered a lot of attention among the beneficial microbes [133] that produce cyclic lipopeptides, amphiphilic molecules with a short peptide chain coupled to the lipid tail, mainly responsible for the biological control impact of pathogens [134,135]. The majority of *Bacillus* species synthesize physiologically active lipopeptides, including *B. subtilis*, *B. velezensis*, *B. megaterium*, *B. amyloliquefaciens*, *B. safensis*, *B. cereus*, and *B. tequilensis*, which devote between five and eight percent of the synthesis of their full genome to bioactive secondary metabolites, including lipopeptides, bacteriocins and siderophores [136]. According to Qiao et al. [137], the wheat root bacteria *B. subtilis* E1R-j has an inhibitory impact on a variety of plant diseases and can stop yellow rust growth mostly by rupturing germ tubes and urediniospores and releasing protoplasm. The effectiveness of this BCA against the pathogen was examined using three formulations produced by E1R-j, FLBC, FL, and BCS [138,139]. Instead of inducing host resistance, directly inhibiting the impact on the rust pathogen was responsible for the decrease in disease severity shown in the greenhouse trial (Table 1). *B. subtilis* E1R-j, if pre-sprayed or concurrently with Pst inoculation, reduces the severity of the disease, underscoring the significance of the timing of the application of *B. subtilis* products [140,141]. Serenade ASO (*B. subtilis* QST713) is a biofungicide used to lessen the severity of Pst in winter wheat and *Blumeria graminis*, offering moderate control of 20–65%. However, for best results, it must be applied in combination with other products [139]. Matzen et al. [45] describe the mechanism of action of *Bacillus* species such as microbial disruptors of pathogenic cell membranes. According to a few studies, *B. subtilis* may also be able to increase host plant resistance, which might explain some of these effects [141,142,143]. Contrary to chemical treatment, biofungicide’s ability to control yellow rust varies significantly from place to place and year to year [139]. This suggests that *B. subtilis*, as a biofungicide, mostly prevents the disease from developing and only serves as a cure at the very beginning. Additionally, *B. subtilis* inhibits the germination, extension, and penetration of germ tubes in spores [141,144,145]. Mejri et al. [15] conducted in vitro and in planta study to demonstrate the biological activity of three cyclic lipopeptides extracted from *B. subtilis* (Fengycin F, mycosubtilin M, and surfactin S) as well as two mixes (S + M) and (S + M + F) on wheat against STB. When these biomolecules were applied on the foliar part of the wheat varieties “Dinosor” and “Alixan” at a rate of 100 mg L^−1^ two days post-fungal inoculation, the severity of the disease was significantly reduced (disease decrease of up to 82% with S + Mon Dinosor), indicating that rather than acting as biofungicides, these lipopeptides operate on wheat against *Z. tritici* to promote resistance [15].

Two *B. velezensis* phyllosphere bacteria isolated from wheat ears, S1 and S6, and their cell-free culture filtrates indicated considerable antifungal activity against STB in vitro. For the culture filtrate, the semi-maximal and minimal inhibitory dilutions were 1.4% and 3.7%, respectively, for strain S1, and 7.4% and 15% for strain S6. Both strains generated cyclic lipopeptides from several families, but only strain S1 produced bacillomycin D, according to a MALDI-ToF study [136]. Natural elicitors, which are now among the most promising BCAs, produce plant resistance to a wide range of diseases. To boost wheat’s defenses against the challenging fungus *Z. tritici*, this study focuses on the eliciting qualities of cyclic surfactin lipopeptide [15,146,147]. Surfactin from *B. amyloliquefaciens* S499 was tested in greenhouse tests for its ability to defend against *Z. tritici*, similar to the chemical elicitor reference Bion^®^ 50WG, which provided 70% protection for wheat against it [148]. When the cell-free culture filtrate or cell suspension of B. subtilis TE3 was applied under extremely optimal conditions (100% relative humidity and 28 °C), there was a significant reduction in *B. sorokiniana* (98%) [149]. The extract and culture filtrate of *B. subtilis* XZ16-1 also exhibited significant inhibition of *B. graminis* spore germination. In comparison to the chemical fungicide triadimefon, the control effects of 100% culture filtrate on *B. graminis* in wheat were 81% and 83%, respectively [150]. In vitro, *B. velezensis* BZR 517 and BZR 336 g had an antagonistic effect against *P. tritici-repentis*, inhibiting mycelial growth by 72 to 94% and inducing degenerative changes [57]. In pot trials, spraying and seed treatment with a bacterial suspension of *B. subtilis* BJ-1 suppressed rice blast by more than 50% [151]; other *Bacillus* strains have shown antagonistic activity against rice blast, such as *Bacillus methylotrophicus* BC79, *Bacillus safensis* B21, *B. tequilensis* JN-369, *B. cereus* YN917, and *B. velezensis* ZW10 [152,153,154,155,156].

#### 3.2.2. Biocontrol with *Pseudomonas* spp. as BCAs

*Pseudomonas* spp. produces a wide range of secondary metabolites, including antibiotics such as phenazines, diacetyl phloroglucinol, and hydrogen cyanide (HCN). The production of phenazines demonstrates the potential for use as a BCA against numerous diseases affecting various crops. Phenazine-1-carboxylic acid (PCA), phenazine-1-carboxamide (PCN), pyocyanin (PYO), and hydroxy phenazines (OH-PHZ) are directly involved in reducing several diseases caused by bacteria, fungi, and oomycetes [157,158]. In vitro, *P. putida* BK8661 inhibited *S. tritici* and *P. recondita* f. sp. *tritici* on wheat leaves, achieving reductions of 71% and 94%, respectively. *P. putida* BK8661 produces antibiotics, siderophores, and minor quantities of hydrogen cyanide. Pleiotropic mutants of strain BK8661 lacking the ability to synthesize siderophores and antibiotics were less successful in symptom suppression than the wild type. By integrating the hcnABC gene cluster from *P. fluorescens* CHAO, pleiotropic mutants were derived, resulting in HCN-overproducing derivatives. In the absence of antibiotics and siderophores, overproduction of HCN by this bacterium resulted in a minor but statistically significant increase in the suppression of symptoms caused by *P. recondita* and *Z. tritici* on wheat seedling leaves [128,159]. Similarly, other plant pathogenic bacteria and fungi were suppressed in vitro by *P. fluorescens* PFM2, which also attacks *S. tritici*. The liquid glycerol–peptone–phosphate medium, in which strain PFM2 was cultured for four weeks, yielded three inhibitory chemicals. The most abundantly recovered chemical (70 mg L^−1^) was 2,4-diacetyl phloroglucinol [160]. When applied to wheat seedlings three hours before the pathogens inoculation, soil-isolated *Pseudomonas aeruginosa* LEC 1 reduced STB by 88% and brown rust by 98%. Antibiotics phenazine-1-hydroxyphenazine (phOH) and chlororaphine were produced by the fractionation and characterization of two inhibitory substances obtained from cultures of *P. aeruginosa* LEC 1 on thin-layer chromatography plates. In wheat seedlings, 160 mg L^−1^ phOH reduced STB by 61% and brown rust by 75% [161]. It also increased the cellular activity of one of the three isozymes, catalase, the only superoxide dismutase produced by *S. tritici*, while decreasing the activity of peroxidase. The potential for the proteolysis of fungal proteins is examined concerning the synergy between the phenazine derivative and the protease in the development of oxidative stress [162] (Table 1).

#### 3.2.3. Other Antagonistic Bacteria as BCAs

Although *Streptomyces* spp. rarely live on plant leaves, it has been demonstrated that foliar spraying these organisms’ bioactive compounds can reduce the symptoms of microbial infections [163]. In a recent study, *Streptomyces tauricus* XF, which can colonize the surface of leaves, was isolated from the rhizospheric soil of a peony. It had a biocontrol effect on the rust disease by applying a fermentation filtrate, which resulted in a lethality rate of 61.47% and inhibited the germination of urediniospores by up to 99% by causing the release of their cytoplasm and the deformation of the germ tube [164]. The generation and germination rate of urediniospores were decreased by the hyperparasite *Cladosporium cladosporioides* isolated from *Pst* taupe-colored uredinia, and the color of the uredinia changed from yellow to taupe [165]. In another study, the inhibitory potential of the endophyte *Epicoccum nigrum* HE20, isolated from native wheat, was tested in vitro and showed high suppression (96%) of yellow rust uredospore germination. GC-MS analysis of the *E. nigrum* HE20 filtrate showed the production of substances with an antifungal effect such as lactic acid, butyric acid, hexanoic acid, α-linolenic acid, 10,12-tricosadiynoic acid, and pentadecanoic acid. The use of this bacterium also promoted growth, and wheat leaves have more overall photosynthetic pigments as well [165] (Table 1).

#### 3.2.4. Biocontrol with *Trichoderma* spp. as BCAs

Eighteen strains of fungi (*Trichoderma* spp.) were selected for their significant potential in controlling *Septoria* species. This potential could be explained by the different modes of action of these antagonists, including competition, mycoparasitism, and antibiosis, as described previously [166,167,168,169,170,171]. In vitro experiments carried out by Villa-Rodriguez et al. [172] showed that the chitinolytic system of strain TSM39 reacts to *B. sorokiniana* via a signaling pathway, which could have potential biotechnological applies to enhance its biocontrol activity. *Trichoderma atroviride* (SG3403) and *Trichoderma harzianum* (SH2303) have shown strong biocontrol activity against the SCLB pathogen [96,173]. Seventy-three *Trichoderma* spp. endophytes obtained from maize leaves from central and northeastern Thailand were tested for their potential to control leaf blight disease caused by *Exserohilum turcicum*. In the field, an application of the fresh formulation of *T. harzianum* KUFA0710 indicated the highest disease reduction at 56%, while the dry formulation caused 47% disease reduction, followed by *T. harzianum* KUFA0713 at 50% and 44% disease reduction when applied with these two formulations, respectively [174] (Table 1).

**Table 1 plants-12-04162-t001:** Main beneficial bacteria against leaf diseases for cereal crops.

Antagonist Strains	Origin	Target Organism Pathogen	Targeted Crop	Results	Antifungal Metabolites/Mode of Action	References
*B. subtilis* BBG13, BBG125, and Bs2504	ProBioGEM, Centre Wallon de Biologie Industrielle	*Zymoseptoria tritici*	Wheat	In vitro and in vivo studies: Mycosubtiline formulations inhibit STB growth, with demi-maximal inhibitory doses of 1.4 mg L^−1^ for M and (M + S) and 4.5 mgL^−1^ for (M + S + F), respectively.	Mycosubtiline, surfactin, fengycin bacillomycin D	[15]
*B. megaterium* 6A *Paneibacillus xylanexedens* 7A	Yellow rust-resistant wheat	*Puccinia striiformis tritici*	Wheat	In semi-field: Decreased severity by 46.07% and 44.47% for the FLBC effect in curative, while 65.16% and 61.11% in protective effect, respectively.	Antioxidant enzymes: SOD, POD, PPO, and PAL; PR proteins	[145]
*B. subtilis* XZ16-1	_	*Blumeria graminis*	Wheat	Preventive and therapeutic efficacy against powdery mildew was 83.72% and 81.18, respectively.	Solubilize phosphate, fix nitrogen, hydrolases, lipopeptides, siderophores, IAA	[150]
*B. subtilis QST713*	*Serenade^®^ ASO*, *Bayer CropScience*	*Puccinia striiformis*, *Blumeria graminis* spp.	Wheat	In the field: Decreased the severity of stripe rust, offering up to 60% at BBCH development stages 65–69, and powdery mildew with moderate control between 20% and 65%.	Mycoparasitism/Metabolites	[45,139]
*B. subtilis* E1R-j	Wheat roots	*Puccinia striiformis* f. sp. *tritici* (*Pst*)	Wheat	In a condition-controlled greenhouse: Protective mode reduces the severity of disease, and the control efficacy ranged between 54.0% and 87.7%.	Mycoparasitism/Metabolites	[138]
*B. subtilis TE3*	Native Wheat	*Bipolaris sorokiniana*	Wheat	In vivo biological control: Reduced the number of lesions/cm^2^ to 3.06 ± 0.6 and 3.74 ± 0.70 as well as the visual damage to 3–5 and 4–6, respectively.	Chitinase, glucanase; siderophores, indoles, and biosurfactants	[149]
*B. velezensis* (S1, S6)	Wheat ears	*Zymoseptoria tritici*	wheat	Regarding culture filtrates, the minimum inhibitory dilution and the semi-maximum inhibitory dilution were, 15% and 7.4% for strain S6 and 3.7% and 1.4% for strain S1, respectively.	Bacillomycine D	[136]
*B. megaterium* MKB135, *P. fluorescens* MKB21 and MKB91	Barley leaves and grain, oat chaff, and wheat rhizospheres	*Zymoseptoria tritici*	wheat	STB development was postponed (by up to 80%).	Mycoparasitism/Metabolites	[175]
*B. amyloliquefaciens* S499	Rhizosphere	*Zymoseptoria tritici*	Wheat	Surfactin provided wheat with a 70% defense against *Z. tritici* in greenhouse tests.	Surfactin, SA and JA signaling pathways	[148]
*B. velezensis* BZR 517 and BZR 336 g	Rhizosphere of winter wheat	*Pyrenophora tritcii repentis*	Wheat	In vitro: induced degenerative alterations in mycelium and decreased its development by 72.4–94.3%. In a three-year field study, BZR 517 and BZR 336 g increased yield by 5.0–7.6%.	Mycoparasitism/Metabolites	[57]
*Pseudomonas putida* BK8661	Wheat leaves	*Zymoseptoria tritici* and *Puccinia recondita* f. sp. *tritici*	Wheat	On wheat leaves, *Septoria tritici* and *Puccinia recondita* f. sp. *tritici* are inhibited from growing in vitro.	Siderophores, antibiotics, HCN	[128,159]
*P. fluorescens* PFM2	Wheat phyllosphere	*Zymoseptoria tritici*	Wheat	In vitro: After 3, 7, and 14 days, the zone of inhibition increased from 0 to 9 cm, 1 to 6, and 1 to 9 cm, respectively.	2,4-diacetylphloroglucinol	[160]
*Pseudomonas aeruginosa* LEC 1	Soil	*Zymoseptoria tritici*	Wheat	Inhibit *Septoria tritici* by 88% and *Puccinia recondita* by 98% when applied to wheat seedlings 3 h before inoculation with the pathogens.	1-hyroxyphénazine (phOH), catalase	[161]
*T. harzianum sensu lato* TSM39	Soil	*Bipolaris sorokoniata*	Wheat	Cellular elements of *B. sorokiniana* stimulate the chitinolytic system of strain TSM39.	Mycoparasitism	[172]
*Streptomyces tauricus* XF	Rhizospheric soil of peony	*Puccinia striiformis*	wheat	The control effects of FL and AC reached 68.25%, and 65.48%, respectively, in the greenhouse. Using XF fermentation broth, yellow rust disease indices were considerably decreased by 53.83%. in the field.	ROS, (PAL), β-1,3-endoglucanases, chitinases, endochitinase, and peroxidase	[164]
*Cladosporium cladosporioides* R23Bo	*Puccinia striiformis*	*Puccinia striiformis*	Wheat	Reduce the urediospore germination rate. The color of uredinia went from yellow to taupe.	Hyperparasitism	[165]
*Epicoccum nigrum* HE20	Healthy wheat	*Puccinia striiformis*	Wheat	In the greenhouse: Reduction of severity by 87.5%.	POD, PPO, and CAT, butyric acid, hexanoic acid, α-linolenic acid, lactic acid, pentadecanoic acid, and 10,12-tricosadiynoic acid Defensive genes (JERF3, GLU, and *PR1*)	[176]
*B. subtilis* BJ-1	Contaminated *Magnaporthe oryzae* culture plate	*Magnaporthe oryzae*	Rice	Detached leaves were inhibited by 10^8^ CFUmL^−1^ (BC) or 5% (Fl) of BJ-1.	Surfactin, fengycin, subtilin, and bacilysin. ISR	[153]
*Bacillus safensis* B21	*Osmanthus fragrans* Lour. Fruits	*Magnaortae oryzae*	Rice	Inhibition of hyphal growth.	Iturin A2, A6	[150,177]
*B. tequilensis* JN-369	Rice	*Magnaortae oryzae*	Rice	The efficacy of biocontrol in protective tests and therapeutic tests on detached rice leaves was up to 74.08% and 62.96%, respectively.	Plant growth and resistance induction	[154,177]
*Bacillus cereus* YN917	Rice leaf	*Magnaporthe oryzae*	Rice	The efficacity before and after inoculation was 68.15% 65.61%, respectively, under detached leaf and greenhouse conditions.	IAA, siderophores, protease, ACC deaminase, cellulase, amylase, β-1,3-glucanase, and phosphate solubilization	[155]
*B. subtilis* B47	Tomato	*Bipolaris maydis*	Maize	In the field, the control efficacy increased to 64.2% when iturin A_2_ concentration was raised to 500 mg kg^−1^.	Iturin A2	[97]
*Trichoderma harzianum* SH2303 *T. atroviride* SG3403	Soil	*Cochliobolus heterostrophus*	Maize	In-field and greenhouse conditions: synergistic application difenoconazole-propiconazole (DP). +SH2303 showed 60% of control.	SAR, PAL, CAT, SOD SA pathway (PR1)	[96,173]

## 4. Induction of Cereals Defense Mechanisms

Plants are always under attack from diseases with various life strategies at any time of the year. While some of these pathogens multiply exterior of plant tissue, others can directly enter plant cells. The earliest contact between plants and pathogens occurs in the apoplast, where membrane-based pattern recognition receptors (PRRs) recognize microbial elicitors identified as pathogen-associated molecular patterns (PAMPs) in cereals [178]. Pathogens release many effectors to interfere with cellular processes throughout the infection process. As opposed to PAMPs, effectors are more diverse and can include toxins, proteins, chemical substances, or hormones. They boost the pathogen’s infectious potential by using the pathogen or by reducing the host’s defenses. Defined as nucleotide-binding domains, proteins with leucine-rich repeats (also known as NB-LRRs) are intracellular receptors that detect certain effectors sent into the plant cell to activate effector-triggered immunity (ETI). Recognition of microbial PAMPs via plant PRRs (receptor-like kinases) triggers the first line of defense, known as PAMP-triggered immunity (PTI). The rapid response involves an influx of extracellular Ca^2+^ into the cytoplasm, followed by the induction of a cellular oxidative rupture that generates reactive oxygen species (ROS) and stimulates mitogen-activated protein kinases, among other reactions. Choudhary et al. [179] have accurately induced resistance and its mode of action in plants. Plants can obtain an increased level of resistance to pathogens after being exposed to biotic stimuli provided by different BCA applications and protect against foliar diseases [139,180] by inducing a stable defense state or ISR in plants [142,181]. ISR depends on ethylene and jasmonate-regulated pathways [182]. A network of interconnected signaling pathways regulates induced plant defenses against pathogens, the main plant signaling molecules being salicylic acid (SA), ethylene (ET), jasmonic acid (JA), and probably nitric oxide (NO) [183]. Based on activation, this mechanism of defense genes results in a quicker or more powerful reaction to pathogen attack, and a rapid accumulation of H_2_O_2_ is observed against wheat mildew and rice blast [184], indicating that ISR activates the first steps of plant protection [184]. SAR may be triggered by exposing the plant to avirulent, virulent, or non-pathogenic microbes, in which there is an accumulation of proteins (PR) linked to pathogenesis, such as glucanase, chitinase, and SA [181]. Treated seedlings also demonstrated a higher expression of pathogenesis-related protein genes (PR), β-1,3-endoglucanases (PR-2), antifungal protein (PR-1), endochitinase (PR-4), ribonuclease (PR-10), and peroxidase (PR-9) against *Z. tritici* [148]. This high expression of PR protein genes could be crucial to triggering the host defense mechanism against yellow rust [145]. A study, using *B. cereus* YN917 showed the production of 1-aminocyclopropane-1-carboxylic acid (ACC) deaminase and indole acetic acid (IAA), which is a significant trait of plant growth-promoting microorganisms [155]. The NPR1 protein, a redox-mediated protein utilized as a transcriptional co-activator of PR genes, is one of the essential elements controlling the SAR pathway. A change structure of the protein is required to activate PR genes induced by the SA receptor NPR1 gene [128,185] (Figure 4).

## 5. Advancements in BCAs: Discussion on Screening, Application, and Future Perspectives

BCAs, defined as organisms or microorganisms that combat plant pathogens, play a crucial role in safeguarding cereal crops against diseases such as Tan spot (*Pyrenophora tritici-repentis*), yellow rust (*Puccinica striiformis*), Fusarium ear blight (*Fusarium culmorum*), and STB [186]. This article scrutinizes the expansive realm of BCAs, elucidating their significance in resisting pest populations and averting crop diseases naturally. We explore the diverse array of bacteria (*Bacillus* spp., *Pseudomonas* spp.) and fungi (*Trichoderma* spp.), highlighting their pivotal role in large-scale pest management without causing damage to the main crop [187].

### 5.1. Challenges of Conventional Practices

The conventional use of chemical pesticides, while boosting crop yields, raises significant concerns for human health and ecosystems. Adverse effects include disruptions to the ecological balance, posing risks of muscular or nerve disorders upon exposure, and potential long-term impacts on the immune system, cellular respiration, and skin health [188,189,190]. Environmental pollution and regulatory bans on hazardous pesticides further underscore the need for alternative approaches.

### 5.2. BCAs as Sustainable Alternatives

Researchers are actively developing alternatives to potentially replace the chemical pesticides, with BCAs emerging as a promising solution [188]. Coined by Harry Scott Smith, the term “biocontrol agents” encompasses a broad spectrum of organisms used in entomology and plant pathology to control pests. Microbes, extracts, or fermented products from natural sources also act as BCAs, offering versatile solutions with various effects on target pathogens [191,192].

### 5.3. Field Application and Challenges

Although BCAs are extremely promising, challenges persist in their field application, which could be compromised by climate change scenarios, where increased temperature variations, humidity changes, altered rainfall patterns [193], and other weather conditions may pose challenges to their performance, there were also parameters such as time of application, inoculation technology, spore survival, formulation and storage which need to be rigorously evaluated to optimize efficacy [194].

### 5.4. Biotechnological Insights and Environmental Impact

Advancements in plant and agricultural biotechnology have facilitated the isolation of BCAs, including *T. harzianum*, *T. viride*, and *T. koningii* for controlling pathogens across various crops [195,196,197,198]. However, challenges such as field applicability, cost considerations, and potential environmental impact persist [108,199].

### 5.5. Systemic Resistance and Molecular Insights

BCAs applied to grain surfaces induce systemic resistance against fungal infections, triggering changes in gene expression and enzymatic activities [200,201]. Molecular approaches unveil complex interactions between antagonistic microbes, hosts, and pathogens, providing insights for tailored solutions [202,203].

### 5.6. Commercial Landscape and Future Perspectives

Commercially available BCAs continue to evolve, with an increasing number of species and producers globally. The benefits of BCAs, including specificity and sustainability, position them as a compelling alternative to chemical pesticides. Public acceptance, lower risks to farmworkers, and potentially lower greenhouse gas emissions further underscore their ecological appeal [14,204].

BCAs present a promising avenue for replacing chemical pesticides. As biopesticides, biofertilizers, and plant development stimulators, BCAs offer diverse action mechanisms. However, further research and development investments are crucial, focusing on dosage, formulation, environmental impact, and their effects on native plant microflora. Understanding the genes, gene products, and signaling molecules responsible for antagonistic activity is imperative for creating more effective BCAs. Advances in formulation processes, encapsulation techniques, and mass production methods are essential to unlocking the full potential of these agents for sustainable and effective pest management [205].

Exploring nanotechnology’s integration for enhanced efficiency, beneficial microorganisms could be encapsulated in nanomaterials, enabling targeted delivery and improved stability. Tailoring nanomaterials to enhance adhesion on plant leaves ensures prolonged effectiveness, while controlled release mechanisms offer precision in disease prevention. Incorporating nanosensors allows for early detection, and understanding nanomaterial–plant interactions is crucial for optimizing protective effects. However, ethical considerations and environmental risk assessments will be essential as this nanotechnological approach unfolds [206].

## 6. Conclusions

In this study, the important pathogens causing diseases in wheat, rice, and maize leaves were discussed along with their current control methods by highlighting the potential of biological control with BCAs that is only beginning to emerge. This work demonstrated how beneficial bacteria can be affiliated with plants’ defense against foliar diseases of cereals by summarizing the general knowledge of the metabolic processes involved in interactions between plants and pathogens. Current research focuses on BCA formulations and genetic resistance for combating these diseases. While there have not been many exhaustive studies on this topic, the exploration of biological control techniques has recently gained significant momentum. Biological control is particularly worth considering given the current trend to reduce the amount of pesticide contamination that harms the environment.

## Figures and Tables

**Figure 1 plants-12-04162-f001:**
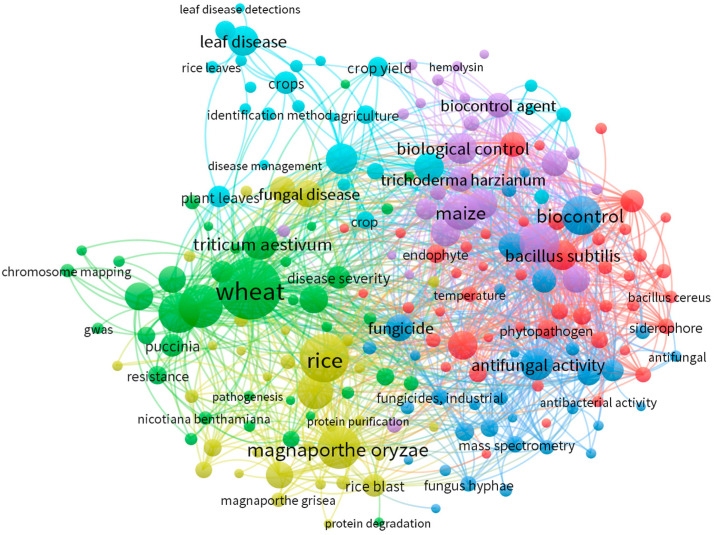
Relevant articles (n = 940) from the Scopus database were subjected to a bibliometric analysis using particular keywords such as “Cereals”, “Wheat”, “Rice”, “Maize”, “Biological Control Agents”, and “Leaf diseases”.

**Figure 3 plants-12-04162-f003:**
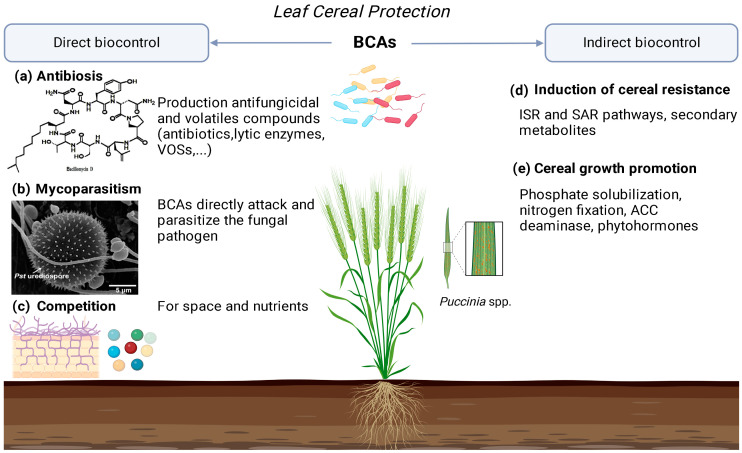
Direct and indirect biocontrol using BCAs as bioprotectants against leaf cereal pathogens.

**Figure 4 plants-12-04162-f004:**
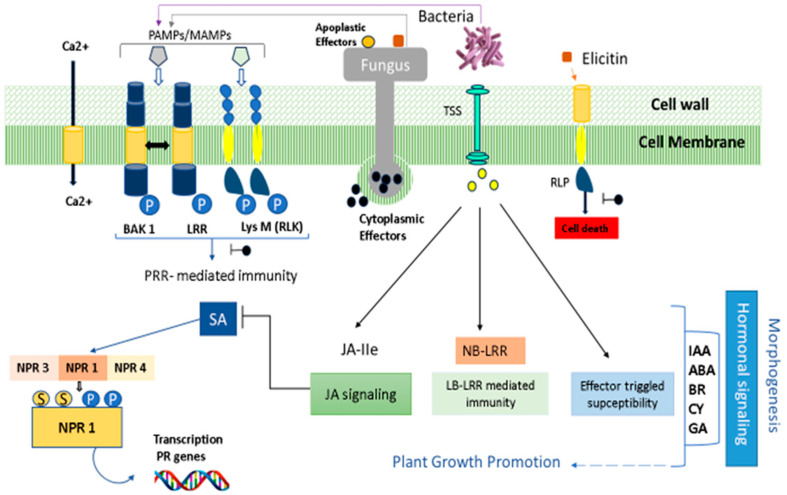
Plant/pathogen interaction involves several metabolic pathways that are implicated in plant immunity to harmful agents. Among these pathways, microbe- or pathogen-associated molecular patterns (MAMPs/PAMPs) are transmitted by pathogen agents to the apoplast as apoplastic effectors, or inside host cells as cytoplasmic effectors, to disrupt plant cell metabolism. MAMPs can be sensed via cell surface receptor proteins (RLPs) or pattern recognition receptors (PRRs; receptor kinases (RLKs) and trigger downstream phosphorylation cascades and cause elevated concentrations of [Ca^2+^] and reactive oxygen species (ROS). Pathogenic effectors are recognized by intracellular receptors, nucleotide-leucine-rich-repeat binding sites (NLRs or NB-LRRs), triggering downstream responses, notably the accumulation of SA. Modulation of gene expression, protein synthesis (PR), and the formation of antimicrobial metabolites are all effects of defense signaling.

## Data Availability

The data used for the analyses in this study are available within the article.

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
