# Peer review of "Beneficial Microorganisms as Bioprotectants against Foliar Diseases of Cereals: A Review"

_plants, 2023, doi:10.3390/plants12244162_

Round 1

Reviewer 1 Report

Comments and Suggestions for Authors

The present article entitled "Beneficial Microorganisms as Bioprotectants against Leaf cereal 2

Diseases: A review"  based on a nice and interesting theme.Authors have tried to compile a review study on the beneficial microorganisms especially Bacillus and Pseudomonas and trichoderma strains and their role in plant protection.

However,author have discussin these section only with their examples of application. Author should also cover the formulation mechanism, delivary methods, what is the challenges in their broad application.

- Currently screening of biocontrol agents and their field application is a challenging task, author should  add a section on this issue

- Format the table, all example of wheat at one place ,and so on

-Add a future perspective section 

Comments on the Quality of English Language

The present article entitled "Beneficial Microorganisms as Bioprotectants against Leaf cereal 2

Diseases: A review"  based on a nice and interesting theme.Authors have tried to compile a review study on the beneficial microorganisms especially Bacillus and Pseudomonas and trichoderma strains and their role in plant protection.

However,author have discussin these section only with their examples of application. Author should also cover the formulation mechanism, delivary methods, what is the challenges in their broad application.

- Currently screening of biocontrol agents and their field application is a challenging task, author should  add a section on this issue

- Format the table, all example of wheat at one place ,and so on

-Add a future perspective section 

Author Response

The present article entitled "Beneficial Microorganisms as Bioprotectants against Leaf cereal Diseases: A review» based on a nice and interesting theme. Authors have tried to compile a review study on the beneficial microorganisms especially Bacillus and Pseudomonas and Trichoderma strains and their role in plant protection. However, author have discussion this section only with their examples of application. Author should also cover the formulation mechanism, delivary methods, what is the challenges in their broad application.

Comment: Currently screening of biocontrol agents and their field application is a challenging task, author should add a section on this issue

Answer: We express our gratitude to the reviewer for their insightful comment. We have added the section in the manuscript.

Comment: Format the table, all example of wheat at one place, and so on

Answer: The table is reorganized. Please see the revised paper.

Comment: Add a future perspective section 

Answer: the section was added. Please see the revised MS.

Reviewer 2 Report

Comments and Suggestions for Authors

Dear Authors,

A manuscript „Beneficial Microorganisms as Bioprotectants against Leaf cereal Diseases: A review“ was submitted by Ilham Dehbi et al., as Review to Plants journal (Manuscript ID: plants-2727972). Authors describe fungal pathogens of cereal plants and damage they cause and ways of control focusing on beneficial microorganisms. The review is very useful. Everything looks good except References. The manuscript could be accepted after revision of References.

Minor revision

Please check your reference list one more time very carefully. For example:

Line 582. Reference 13 - management, S.O.-O. on pest; 2015, undefined An Analysis of the Biopesticide Market Now and Where It Is Going. – Please check, it is not understandable. Authors?

Line 683. Reference 58 - microbiology, T.S.-M.; 2005, undefined Bacillus Subtilis Antibiotics: Structures, Syntheses and Specific Functions. Wiley – please check it. Author?

Line 707. Reference 67 - Russia. Agron. 2022, Vol. 12, Page 373 2022, 12, 373, - Volume and page are written twice?

Please check References 115, 124, 127, 135, 157, 158, 163, 164, 165, 166, 171, 185 (doi), 193.

Latin names of plants and microorganisms must be written in Italic (everywhere in References).

Sincerely, 2023 Nov 15

Author Response

A manuscript „Beneficial Microorganisms as Bioprotectants against Leaf cereal Diseases: A review “was submitted by Ilham Dehbi et al., as Review to Plants journal (Manuscript ID: plants-2727972). Authors describe fungal pathogens of cereal plants and damage they cause and ways of control focusing on beneficial microorganisms. The review is very useful. Everything looks good except References. The manuscript could be accepted after revision of References.

Minor revision

Please check your reference list one more time very carefully. For example:

Comment: Line 582. Reference 13 - management, S.O.-O. on pest; 2015, undefined An Analysis of the Biopesticide Market Now and Where It Is Going. – Please check, it is not understandable. Authors?

Answer: Line 582. Reference 13 was rectified.

Comment: Line 683. Reference 58 - microbiology, T.S.-M.; 2005, undefined Bacillus Subtilis Antibiotics: Structures, Syntheses and Specific Functions. Wiley – please check it. Author?

Answer: Line 683. Reference 58 was rectified.

Comment: Line 707. Reference 67 - Russia. Agron. 2022, Vol. 12, Page 373 2022, 12, 373, - Volume and page are written twice?

Answer:  Line 707. Reference 67- was checked.

Comment: Please check References 115, 124, 127, 135, 157, 158, 163, 164, 165, 166, 171, 185 (doi), 193.

Answer: References have been corrected. Please see the revised paper.

Comment: Latin names of plants and microorganisms must be written in Italic (everywhere in References).

Answer: the scientific names of species written in Italic was added.

Reviewer 3 Report

Comments and Suggestions for Authors

See included.

Comments on the Quality of English Language

Overall, the English language need to be checked.

Author Response

Comment: heading: uncertain if leaf cereal diseases is the right term?

Should it not be cereal leaf diseases or Foliar disease of cereals?

Answer: the term was changed to Foliar disease of cereals.

Comment: The paper tries to cover 3 main crops. I will recommend to reduce it to just cover wheat. The diseases in rice and maize are only described to a minor degree compared with the wheat diseases and the BCA cases are few and do not seem strong in these two crops either. In the text with the BCA very little involves the rice and maize.

Comment: overall, the English language need to be checked.

Answer: the English language was checked.

Abstract

Comment: Line 29-30: rephrase the sentence to. ……. most of which are known to be plant growth promoting rhizobacteria of plants.

Answer: the sentence was rephrased.

Introduction

Comment: Line51: To Include the major groups in the bracket- include ascomycetes possibly delete oomycetes.

Answer: oomycetes were remplaced by ascomycetes.

Comment: Line 54. The sentence is not true for many fungi. Most cause necrotic death and senescence of leaves cells (e.g., Zymoseptoria tritici, Helminthosporium spp etc.).

Answer: The sentence was corrected in MS.

Comment: Line 60-64: the sentences need to be rephrased. It is unclear what the authors really want to say!!

Answer: Line 60-64: the sentences were rephrased. Please see the revised paper.

Comment: Line 59: instead of writing fungicide resistant crop diseases I would replace it with fungicide resistant plant pathogens.

Answer: Line 59: it was replaced.

Comment: Line 70: A new theme is introduced “grass foliar diseases” I recommend to stay with cereal foliar diseases.

Answer: well done.

Comment: Line 82: replace”leaf cereals” with “cereal leaves”

Answer: it was replaced.

Main foliar diseases

The descriptions of diseases are very inhomogeneous. It is very detailed on the two rust diseases and powdery mildew but less detailed for other diseases like Septoria tritici blotch, rice diseases and maize diseases. Even between leaf rust and yellow rust; the way of describing the two diseases are not very similar. For some diseases control measures are included, but for others this is not the case (e.g. rust diseases). You need to have a common way of describing the diseases in order to ensure that the same aspects of each of the diseases are covered. Recommend that the whole section is rewritten and restructured.

Comment: Line 100: loss levels are different from losses given in line 104. Need some consistency.

Answer: Loss levels was modified.

Comments: Line 104-107: The sentence needs to be rephrased.

Answer: The sentence was rephrased.

Comment: Line 113-114: sentence about telia phase should be placed in a different part. It seems out of place.

Answer: sentence about telia phase was replaced.

Comment: Could be placed after line 128 given a short description on how it looks, etc.

Answer: well done.

Comment: Line 129-131: drop the historic description – too many details.

Answer: it was done.

Comment: Line 171: Not right to start the sentence with Even though……. Should be rephrased with e.g. As most wheat varieties lack significant………

Answer: it was rephrased.

Comment: Line 176: Reference 50 deals with powdery mildew not STB.

Answer: it was settled.

Comment: Line 188-189: Sentence does not make sense as a stand alone. Rephrase to something like.

Answer: the sentence was rephrased.

Comment: The epidemic development of powdery mildew is highly influenced by the cultivars resistance and the impact from applied fungicides. Line 197-199: The sentence needs to be rephrased. Does not make sense as it reads now.

Answer: the sentence was rephrased.

Comment: Line200-202. This sentence is a general point which covers all diseases. Should not be stated here as a stand alone, could be moved to section with BCAs.

Answer 20: this sentence was deleted.

Comment: Line 207: Only mention latin name of crop and diseases once. At the first time it is mentioned.

Answer: It was done.

Comment: 213-214: The sentence needs to be rephrased. Once heading should be removed and heading included at the end of the sentence.

Answer: the sentence was rephrased and replaced.

Comment: Line 215: Plant detritus change to Plant debris

Answer: it was changed

Comment: Line 218-219: Add Australia to list of important countries. Why do you now call it bronze spot? This has not previously been introduced.

Answer: Australia was added and bronze spot was rectified.

Comment: Reference 64. Is about triticale not wheat.

Answer: Yes, it was corrected.

Comment: Line 235-244: Very brief description. Gives a very unbalanced element in the paper. Not sure if it makes sense to keep Rice as part of the paper? The rice cases given in table 1 are adding limited to the content of the paper.

Answer: It is very important to mention rice in this manuscript, giving the basics on foliage diseases and the new strains used for biological control, which will probably contribute to future research into other fungal plant diseases.

Comments: Line 246-279: again, the description is very inhomogeneous and give again an unbalanced element in the paper. Not sure if it makes sense to keep Maize as part of the paper. The maize cases given in table 1 are adding limited to the content of the paper.

Answer: in this research we have carried out a bibliometric analysis which has guided us to talk about this subject with the 3 crops, and it is very important to mention maize in this manuscript, giving the basics on foliage diseases and the new strains used for biological control, which will probably contribute to future research into other fungal plant diseases.

Comment: Figure 1: Pictures of diseases are general and some of poor quality (e.g., A). The C picture is from APS home page. Pictures should be acknowledged.

Answer: concerning the photos we are working on in a vague way, and the quality was rectified.

Comment: Line 288: IDM should be DMI

Answer: it was changed.

Comment: Line 290: change to diseases on cereal leaves.

Answer: Line 292-294: Delete the sentence about historical used chemicals.

Comment: 294-299: Mention only the actives not the commercial names, which vary a lot depending on country.

Answer: the mention of the commercial names of the products is interesting because it contains the associations of the active ingredients.

Comments: Line 300: You only mention rust disease. What about the chemical control of all the other diseases?

Answer: We added a few examples on other diseases

Comment: Line 302-304: Again, this is also only addressing rust diseases, what about the other diseases?

Answer: We express our gratitude to the reviewer for their insightful comment. In response, we have incorporated additional examples specifically addressing diverse cereal diseases.

Comment: Line 309: Add use to finish the sentence.

Answer: done. Please see the revised MS.

Comment: Line 312: ISR – Write out completely what it covers – following this you can use abbreviations.

Answer: we thank the reviewer for this comment. Subsequent to this comment, necessary modifications have been implemented in the revised version of the paper.

Comment: Line 312: Adjust! ……. Indirectly by ISR in cereal. BCAs has been widely studied for its potential benefits……….

Answer: Modification was done. Please see the revised paper.

Comment: Line 314: Reference 127, Add more information on the paper.

  1. Fungi 2022, 8(6), 632; https://doi.org/10.3390/jof8060632

Answer: the reference was corrected.

Comment: Biocontrol with Bacillus spp.

The chapter is core for the review. The heading with Bacillus spp is slightly misleading as also other BCA species are mentioned.

Answer: We extend our appreciation to the reviewer for providing valuable feedback. In the interest of specificity, examples of other antagonistic bacteria have been excluded from this section, focusing only on species within the Bacillus genus.

Comment: Line 347: What does the mixes consist of. It does not make sense for new readers, to understand what is behind.

Answer: we thank the reviewer for this comment. The first mixture comprises Surfactin + Mycosubtilin while the second consisted of Surfactin + Mycosubtilin + Fengycin. Please refer to the revised manuscript for these updates.

Comment: Line 359: remove decimals from text. Two decimals highlight an accuracy which does not exist in these studies.

Answer: The modification requested was done.

Comment: Line 372: Remove Pseudomonas from this part for a to later section, if you keep the heading.

Answer: The modification requested was done. Please see the revised MS.

Comment: Line 381-384: not sure I read and understand this sentence right. Please rewrite or reconsider. What level of control did you achieve from the chemical substance and the BCA’s respectively?

Answer: The sentence was rewriting. the level of control obtained for the chemical substance and BCAs is 70% and 83% respectively.

Comment: Line 384-386 Remove decimals.

Answer: The modification requested was done.

Comment: Line 392-417. This section needs a rewriting, as some of the sentence does not make sense.

Answer: Modifications have been made to this section. Please consult the Manuscripts.

Comment: Line 448. SCLB pathogens. What is this?

Answer: The abbreviation 'SCLB' designates Southern Corn Leaf Blight, and its mention can be found in line 269

Comment: Line 452-453: remove decimals.

Answer: the modification was done.

Comment: Line 455-473 Genetic resistance:

This chapter seems out of the context. If information on genetical resistance should be included then it could be included in the section on the specific plant pathogens.

Answer: This chapter was removed.

Comment: Line 490-: This sentence does not make sense.

Answer: the sentence was removed.

Comment: There is no discussion. This is extraordinary for a review!

Miss a critical discussion on the achieved results from testing of BCA. From practice we know that there still is big gap in the control levels. The BCA are much less persistence and sensitive to weather factors.

Answer: We express our gratitude to the reviewer for this insightful comment. In response, a dedicated paragraph has been incorporated to address the sensitivity of Biological Control Agents to environmental factors, elucidating their impact on the efficacy of BCAs as antagonists. Please see the revised paper.

Comment: References 13: this does not make sense. 38 and 39 are both replicates.

Answer: the references was corrected.

Comment: Ref 58: need to be rewritten.

Answer: done.

Round 2

Reviewer 1 Report

Comments and Suggestions for Authors

I  am not be able to see any revision in he article. It seems only authors have rearranged the table.

Comments on the Quality of English Language

I  am not be able to see any revision in he article. It seems only authors have rearranged the table.

Author Response

Comment: I am not be able to see any revision in the article. It seems only authors have rearranged the table.

Answer: We apologize for the inconvenience regarding our revised MS which was not submitted with the track change feature. Please see the latest revised version.

Reviewer 3 Report

Comments and Suggestions for Authors

Comments to new version.

Many elements have been adjusted but there is still a lot of elements, which need to be adjusted. There is still a lot of inconsistency when you read the paper.

Name of pathogens are very inconsistent.

Septoria tritici blotch (STB) is often listed as Septoria.  

e.g. Line 158  Leaf Septoria has to be adjusted to Septoria tritici blotch  or the abbreviation STB. You need to check the whole review to ensure consistence. A lot of cases mentioning Septoria from line 380-468 need to be adjusted.  

The latin name is written as Zymoseptoria tritici or Septoria tritici. It has in all cases to be adjusted to  Zymoseptoria tritici.

The historical description of Puccinia dispersa should be deleted. This is too detailed in this paper, as this should similarly be included for all pathogens to have consistency.

Line 215 Start this paragraph with the sentence from line 219.

Line 309-313. You still need to adjust the chemicals mentioned. Commercial names should be removed as the tradenames are different throughout the world.

Remove the sentence in line 314-316 about Brazil. It is too specific.

Line 317: delete the word methoxiacrulates as this is identical to strobilurins.

Line 467: delete decimals

The new section 6. Has been added in the new version to meet my proposal for including a discussion.

Why not call it Discussion?

This  Section does bring in relevant elements but it brings in citations from many specific papers and does not give a very coherent discussion.

Line 525: Four diseases is mentioned, which are not specified using latin names  or in which crops that they are appear. It appears as the authors has picked different citations and not critically evaluated there relevance for this specific paper.

Line 532 and 340: mention fertilizers, but again this paper is about disease control not about fertilisers.  

Line 559: It is stated : Applied to fruit surfaces induce systemic resistance. Why include fruit in this paper which is all about cereals.

Line 554 mention LK11 out of the blue. The section on Trichoderma 456-469 does not mention this strain. It seems strange to suddenly mention this here.  

The sentence about the environmental benefits from BCA’s versus traditional fungicides is mentioned too many times. although it might be true it is not ok to be over repeating it. (Line 533, 539, 365, 596)

Line 591: you state : Current research focuses on chemical approaches. Not sure that this is right. More research is currently going into e.g. BCA investigations.  

Lack some wider discussion on the potential use of BCA's. How much is sprayed in maize, rice and wheat. How much can be replaced. 

The substances should be used preventatively. This means that the farmers might have to spray more times than with chemical fungicides. 

Author Response

Comment: Many elements have been adjusted but there is still a lot of elements, which need to be adjusted. There is still a lot of inconsistency when you read the paper.

Answer: We thank the reviewer for the constructive comments and suggestions. We have revised all the inconsistencies presented in the MS.

Comment: Name of pathogens are very inconsistent. Septoria tritici blotch (STB) is often listed as Septoria.  e.g. Line 158 Leaf Septoria has to be adjusted to Septoria tritici blotch or the abbreviation STB. You need to check the whole review to ensure consistence. A lot of cases mentioning Septoria from line 380-468 need to be adjusted. 

Answer: Leaf Septoria has been accordingly replaced by STB in the revised version.

Comment: The latin name is written as Zymoseptoria tritici or Septoria tritici. It has in all cases to be adjusted to  Zymoseptoria tritici.

Answer: Done.

Comment: The historical description of Puccinia dispersa should be deleted. This is too detailed in this paper, as this should similarly be included for all pathogens to have consistency.

Answer: The historical description has been deleted and the whole related paragraph was effectively summarized.

Comment: Line 215 Start this paragraph with the sentence from line 219.

Answer: Done.

Comment: Line 309-313. You still need to adjust the chemicals mentioned. Commercial names should be removed as the tradenames are different throughout the world.

Answer: Commercial names have been removed.

Comment: Remove the sentence in line 314-316 about Brazil. It is too specific.

Answer: the sentence has been removed.

Comment: Line 317: delete the word methoxiacrulates as this is identical to strobilurins.

Answer: the word methoxiacrulates was deleted.

Comment: Line 467: delete decimals

Answer: Done.

Comment: The new section 6. Has been added in the new version to meet my proposal for including a discussion. Why not call it Discussion? This Section does bring in relevant elements but it brings in citations from many specific papers and does not give a very coherent discussion.

Answer: We renamed the section to “Advancements in BCAs: Discussion on Screening, Application, and Future Perspectives”.

Comment: Line 525: Four diseases is mentioned, which are not specified using latin names or in which crops that they are appear. It appears as the authors has picked different citations and not critically evaluated their relevance for this specific paper.

Answer: In the revised version, we mentioned four diseases related to cereal crops: Tan spot (Pyrenophora tritici-repentis), yellow rust (Puccinica striiformis), Fusarium ear blight (Fusarium culmorum), and STB.

Comment: Line 532 and 340: mention fertilizers, but again this paper is about disease control not about fertilisers.

Answer: ‘Fertilizers’ are deleted from the subsection. We only kept chemical pesticides.

Comment: Line 559: It is stated: Applied to fruit surfaces induce systemic resistance. Why include fruit in this paper which is all about cereals.

Answer: the ‘fruit’ was replaced with ‘grain’ as it is related to cereals.

Comment: Line 554 mention LK11 out of the blue. The section on Trichoderma 456-469 does not mention this strain. It seems strange to suddenly mention this here. 

Answer: The LK11 strain is deleted from the subsection.

Comment: The sentence about the environmental benefits from BCAs versus traditional fungicides is mentioned too many times. although it might be true it is not ok to be over repeating it. (Line 533, 539, 365, 596)

Answer: We have accordingly reduced the repeating rate of the sentence.

Comment: Line 591: you state: Current research focuses on chemical approaches. Not sure that this is right. More research is currently going into e.g., BCA investigations. 

Answer: The reviewer is right in this comment. We have accordingly corrected the sentence.

Comment: Lack some wider discussion on the potential use of BCA's. How much is sprayed in maize, rice and wheat. How much can be replaced.

Answer: In this paper, we have discussed the most efficient agents that have good results while fighting against certain harmful agents on cereal crops, but regarding the quantity sprayed and replacement it depends on several factors as types of pests, the area of land plot, crop conditions, effectiveness of specific BCAs...

Comment: The substances should be used preventatively. This means that the farmers might have to spray more times than with chemical fungicides.

Answer: Yes, exactly. Biocontrol agents are used to prevent diseases and provide plants with immunity and resistance to infections. They also need to be sprayed with biological material, which is still the most effective choice compared to toxic chemicals.

Round 3

Reviewer 1 Report

Comments and Suggestions for Authors

Author had made significant changes in the revision. The article can be accepted in the present form.

Comments on the Quality of English Language

English is fine